# Subjective-Aligned Dataset and Metric for Text-to-Video Quality Assessment

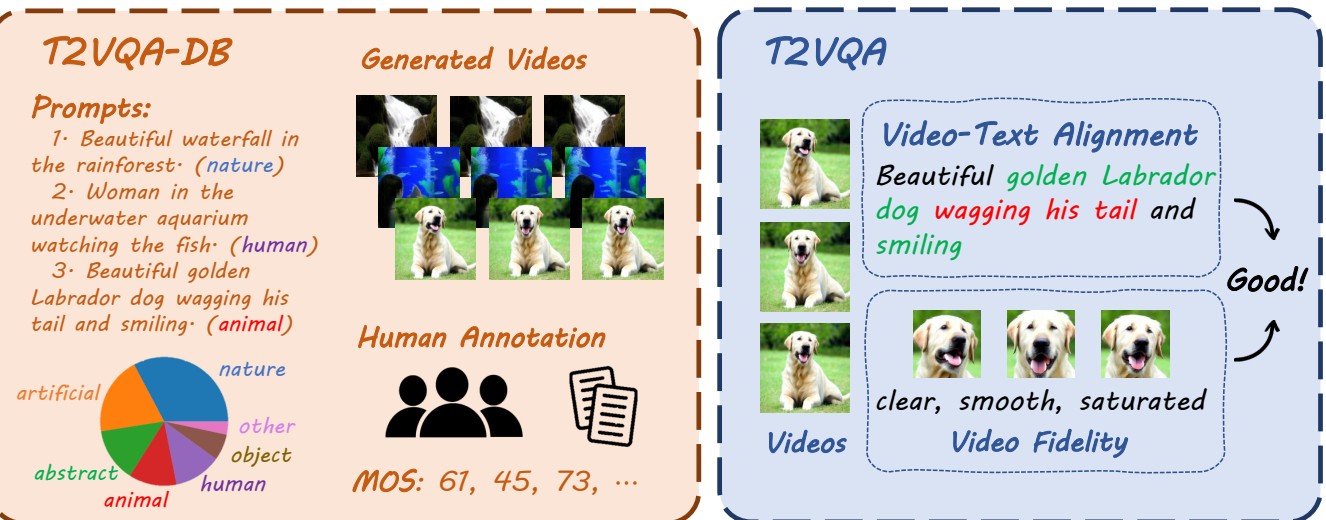

**Figure 1: Overview of the proposed *T2VQA-DB* and *T2VQA*. T2VQA-DB has the largest scale among existing T2V datasets. T2VQA achieves the SOTA performance in evaluating the quality of text-generated videos.**

## ABSTRACT

With the rapid development of generative models, AI-Generated Content (AIGC) has exponentially increased in daily lives. Among them, Text-to-Video (T2V) generation has received widespread attention. Though many T2V models have been released for generating high perceptual quality videos, there is still lack of a method to evaluate the quality of these videos quantitatively. To solve this issue, we establish the largest-scale Text-to-Video Quality Assessment DataBase (**T2VQA-DB**) to date. The dataset is composed of 10,000 videos generated by 9 different T2V models, along with each video's corresponding mean opinion score. Based on T2VQA-DB, we propose a novel transformer-based model for subjective-aligned Text-to-Video Quality Assessment (**T2VQA**). The model extracts features from text-video alignment and video fidelity perspectives, then it leverages the ability of a large language model to give the prediction score. Experimental results show that T2VQA outperforms existing T2V metrics and SOTA video quality assessment models. Quantitative analysis indicates that T2VQA is capable of giving subjective-align predictions, validating its effectiveness. The dataset and code will be released upon publication.

**Unpublished working draft. Not for distribution.**

## CCS CONCEPTS

• **Computing methodologies** → **Modeling methodologies**.

## KEYWORDS

Text-to-video dataset, Video quality assessment , Text-to-video generation

## 1 INTRODUCTION

Video generation, or video synthesis, has been fully developed in the past few years. Text-to-Video (T2V) generation is one of the most studied fields, where a user provides a text description as the guidance for video generation. With the thriving of diffusion-based models, high-fidelity videos can be generated. However, the quality of text-generated videos is diverse which affects the experience quality of subjects. Therefore, a subjective-aligned quality assessment method for them is needed. Unfortunately, existing Video Quality Assessment (VQA) models are unable to accomplish the task well. On the one hand, distortions brought by the T2V generation models, such as the jitter effect, irrational objects, etc, are different from distortions in natural videos. On the other hand, traditional VQA models do not take text-video alignment into consideration, which is a significant evaluation perspective for text-generated videos.

Besides, the most used metrics for T2V generation, such as IS [36], FVD [42], and CLIPSim [46], fail to reflect real user preferences. IS uses the Inception Network [40] to generate a distribution that reflects image/video quality and diversity. It has been criticized for

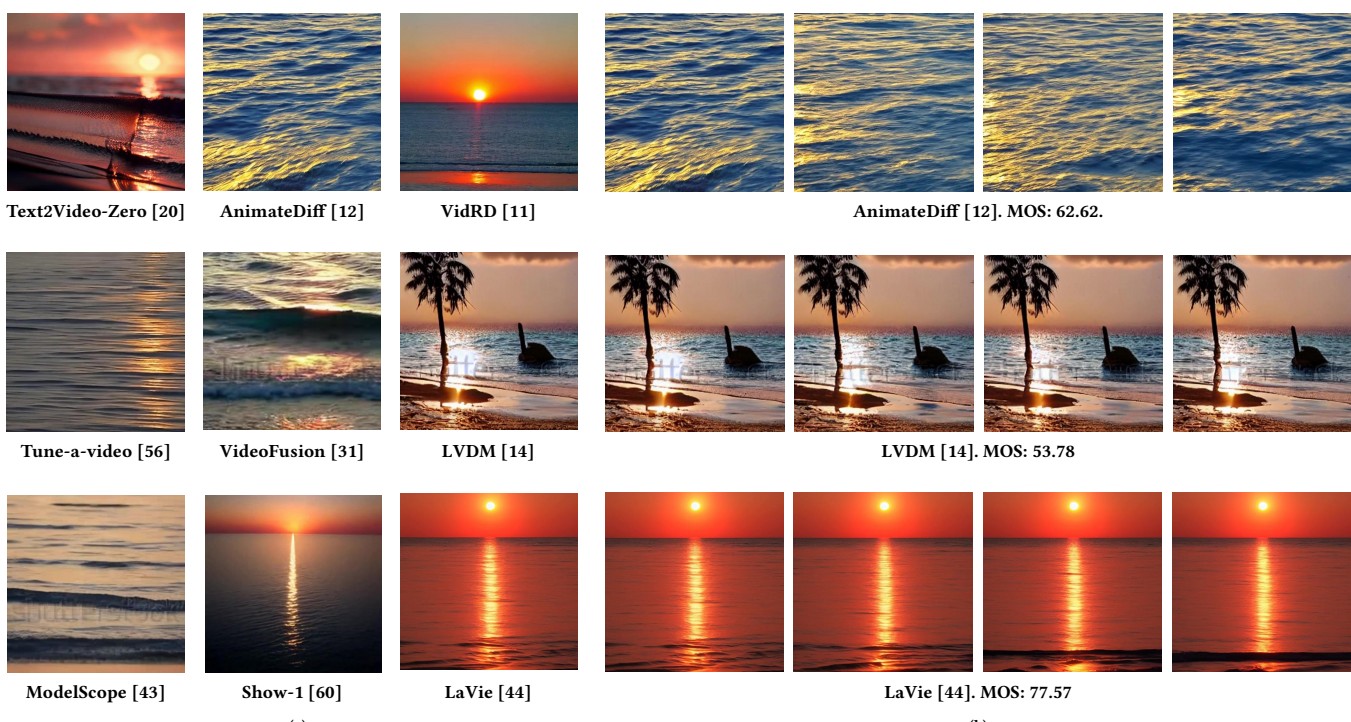

**Figure 2: Video examples generated by prompt: *Sunset on the sea.* (a) Overview of video frames generated by 9 models. (b) Videos generated by AnimateDiff [12], LVDM [14], LaVie [44], and their MOSs.**

its inability to evaluate image/video quality precisely. FVD compares the I3D feature [4] distributions of the generated and natural video pair. The drawback of FVD is that obtaining the reference natural video is usually impractical. CLIPSim takes advantage of CLIP [32] to calculate the similarity between the original text and the generated video content. However, it only considers text-video alignment from the image level, excluding the temporal information and perceptual video quality.

To facilitate the development of a more comprehensive and accurate metric, we establish the largest-scale subjective T2V dataset to date, named Text-to-Video Quality Assessment DataBase (***T2VQA-DB***). The dataset contains 10,000 videos generated by 9 different T2V models using 1,000 text prompts. We also collect each video's Mean Opinion Score (MOS) by conducting a subjective experiment, where 27 subjects score the overall quality of the generated videos. Fig. 2 shows video examples from T2VQA-DB. We anticipate that the T2VQA-DB will benefit the training and testing of subsequent models.

Based on T2VQA-DB, we propose a novel model equipped with multi-modality foundation models for better Text-to-Video Quality Assessment (***T2VQA***). The model utilizes BLIP [27] and Video Swin Transformer (Swin-T) [30] to extract features from text-video alignment and video fidelity perspectives respectively. The features are fused through a cross-attention module, then they are fed into a frozen Large Language Model (LLM) to regress the predicted score. We train and test T2VQA as well as other VQA models on T2VQA-DB. Experimental results show that T2VQA outperforms

existing T2V generation metrics and state-of-the-art VQA models, validating its effectiveness in measuring the perceptual quality of text-generated videos.

We summarize our contributions as follows:

(1) We establish the T2V dataset with *the largest scale to date*, named T2VQA-DB, which includes 10,000 text-generated video sequences and their corresponding MOSs gained from 27 subjects.

(2) We propose a novel transformer-based model for better evaluating the quality of text-generated videos, named T2VQA. The model dissolves the problem from *text-video alignment and video fidelity perspectives*, and then it leverages the ability of an LLM to give a subjective-aligned prediction of the video quality.

(3) The proposed T2VQA outperforms existing metrics for T2V generation and SOTA VQA models on T2VQA-DB and Sora [3] videos, indicating the effectiveness of T2VQA. Qualitative experiments show that T2VQA can benefit in measuring the performance of T2V generation algorithms, giving it practical application prospects.

## 2 RELATED WORKS

### 2.1 Text-to-video Dataset

To the best of our knowledge, only a few T2V datasets have been proposed. They mainly have the following two issues: (1) Insufficient scale: Chivileva *et al.* [5] proposes a dataset with 1,005 videos

**Table 1: Summary of T2VQA-DB and existing T2V datasets. [Keys: Bold: the best].**

| Name | Videos | Prompts | Models | Annotators |
|---|---|---|---|---|
| Chivileva's [5] | 1,005 | 201 | 5 | 24 |
| EvalCrafter [28] | 3,500 | 500 | 7 | 3 |
| VBench [16] | 6,984 | **1,746** | 4 | - |
| FETV [29] | 2,476 | 619 | 4 | 3 |
| **T2VQA-DB (ours)** | **10,000** | 1,000 | **9** | 27 |

generated by 5 T2V models following [25]. EvalCrafter [28] builds a dataset using 500 prompts and 7 T2V models, resulting in 3,500 videos in total. Similarly, FETV [29] is composed of 2,476 videos generated by 619 prompts, and 4 T2V models. VBench [16] has a larger scale with in total of ~1.7k prompts and 4 T2V models. Such scales are not sufficient for the training of deep learning-based models, and cannot comprehensively represent current T2V algorithms. (2) Limited human annotation: ITU-standard [17] requires at least 15 human annotators for subjective study. The dataset proposed by Chivileva et al. [5] is the only one that meets the standard with 24 annotators involved. Both EvalCrafter and FETV only have 3 users for annotation. VBench [16] does not specify the number of human annotators explicitly.

## 2.2 Metrics for T2V Generation

IS [36] and FVD [42] are the two most commonly used metrics for evaluating the quality of generated videos. IS uses the Inception feature to present both image/video quality. FVD measures the distance between the generated video and the natural video. However, both metrics are criticized for poor correlation with human visual perception. CLIPSim [46] measures the text-video alignment by using CLIP [32]. After measuring the similarity between the text and each video frame, it averages them to get the final score. As a result, it only evaluates videos from the image level, losing the information from the temporal domain. As well as it doesn't consider the video quality.

Though many works targeting the evaluation of text-generated images have been proposed [21, 24, 25, 57, 58, 62], only a few metrics tailored for text-generated video evaluation have been proposed. Among them, ViCLIP [45] is a CLIP-based metric for measuring text-video alignment. Chivileva et al. [5] proposes an ensemble video quality metric that integrates text similarity and naturalness. EvalCrafter [28] and VBench [16] build benchmarks to evaluate text-generated video from 18 and 16 objective metrics respectively. FETV [29] and T2V-Score [55] both propose separate metrics for text-video alignment and video quality, without an overall perceptual score for text-generated videos. There is still lack of a simple and effective metric to evaluate the quality of text-generated videos accurately.

Besides the aforementioned metrics, VQA models can also be used for the evaluation of text-generated videos. BVQA [23] transfers knowledge from Image Quality Assessment (IQA) databases and then trains on the target VQA database. SimpleVQA [39] extracts spatial and motion features to regress to the final score. FAST-VQA [47] proposes "fragments" as a novel sampling strategy and the

Fragment Attention Network (FANet) to accommodate fragments as inputs. [8, 9, 22, 63] are works for the evaluation of enhanced videos and digital humans. DOVER [50] proposes to view the quality assessment problem from the technical perspective and the aesthetic perspective, while BVQI [48] and MaxVQA [49] integrates text prompts (e.g., good, bad) into VQA. With the development of LLMs and Multi-modal Large Language Models (MLLM), researchers have started to leverage the advantages of MLLMs to solve VQA problems, as they have trained on massive data. Q-Bench [51] proves that MLLMs can address preliminary low-level visual tasks. Q-Align [53] proposes to train MLLMs using text-defined levels (e.g., fine, poor, excellent) and achieves SOTA results on both IQA and VQA tasks. Though many VQA models have been proposed, they are originally designed for natural videos and do not consider text-video alignment. Therefore, their performance will deteriorate when evaluating generated videos.

## 2.3 Text-to-video Generation

T2V generation refers to a form of conditional video generation, where text descriptions are used as conditioning inputs to generate high-fidelity videos. A common practice is to extend pre-trained Text-to-Image (T2I) models with temporal modules. CogVideo [15] is based on CogView2 [7] and proposes a multi-frame-rate hierarchical training strategy to better align text-video clips. Make-a-video [37] adds effective spatial-temporal modules on a diffusion-based T2I model (i.e., DALLE-2 [33]). VideoFusion [31] also leverages the DALLE-2 and presents a decomposed diffusion process. LVDM [14], Text2Video-Zero [20], Tune-A-Video [56], Animate-Diff [12], Video LDM [2], MagicVideo[66], ModelScope [43], and VidRD [11] are models that inherit the success of Stable Diffusion (SD) [35] for video generation. Show-1 [60] integrates both pixel-based and latent-based text-to-Video Diffusion Models (VDMs). LaVie [44] extends the original transformer block in SD to a spatio-temporal transformer. Recently, OpenAI releases Sora [3], a T2V model that is capable of generating 60s high-fidelity videos, considered as a game changer in T2V generation.

## 3 SUBJECTIVE-ALIGNED TEXT-TO-VIDEO DATASET

As shown in Tab. 1, the existing T2V datasets have a relatively small number of videos. The small scale is not sufficient to represent the diverse performance of T2V generation models, resulting in unreliable quality assessment metrics. Consequently, we propose a Text-to-Video Quality Assessment DataBase, named T2VQA-DB, including 10,000 videos generated by 9 different T2V models. 1,000 prompts are used and 27 subjects are invited to obtain the MOS of each video. In this section, we will describe the establishment of T2VQA-DB and the subjective experiment conducted on it.

## 3.1 Prompt Selection

To guarantee the diversity of the dataset, the prompts used for T2V generation should cover as many aspects as possible. Following [58], we use the same graph-based algorithm from [38] for prompt selection. We first randomly sample 1 million prompts from WebVid-10M [1], which contains 10 million video-text pairs scraped from the stock footage sites. Each prompt is encoded to a vector

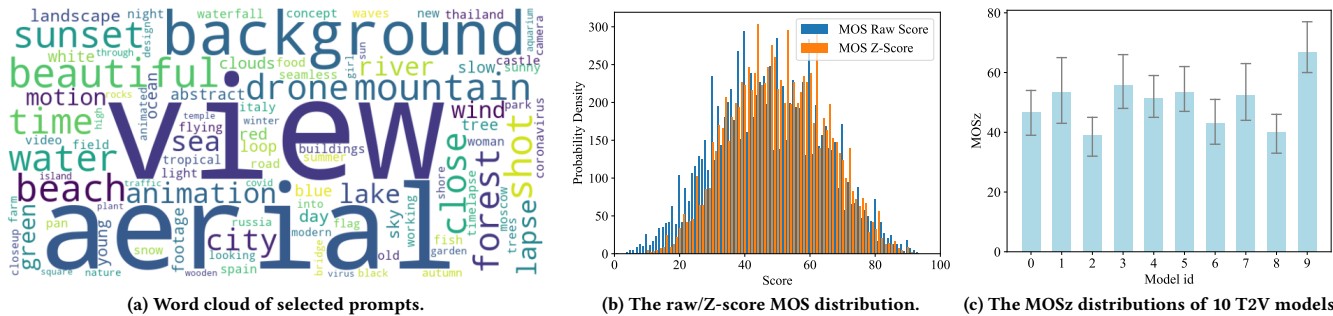

(a) Word cloud of selected prompts.     (b) The raw/Z-score MOS distribution.     (c) The MOSz distributions of 10 T2V models.

Figure 3: (a) The word cloud of the prompts used in T2QA-DB. (b) The distribution of the raw/Z-score MOS. (c) Z-score MOS distributions of 10 T2V models. The model IDs represent sequentially Text2Video-Zero, Tune-a-video[(1)], VidRD, ModelScope, VideoFusion, LVDM, Show-1, Tune-a-video[(2)], LaVie.

representation by Sentence-BERT [34]. The graph-based algorithm integrates them into $k$ groups according to cosine distance. $k$ is a hyper-parameter and we set $k = 100$, resulting roughly 10,000 prompts in each group. Finally, we randomly sample 10 prompts in each group, forming the 1,000 prompts in T2QA-DB. Fig. 3a shows the word cloud of the collected prompts.

### 3.2 Video Generation

We use in total 9 different models for video generation, including Text2Video-Zero [20], AnimateDiff [12], Tune-a-video [56], VidRD [11], VideoFusion [31], ModelScope [43], LVDM [14], Show-1 [60], and LaVie [44]. For Tune-a-video, we utilize two different pre-trained weights, resulting in a total of 10 models for generation. Compared to other T2V datasets, we utilize current advanced T2V generation models as much as possible, making T2QA-DB more representative. Since the default resolution, video length, and frame rate are different in each model, we unify the video format as $512 \times 512$, 16 frames, and 4fps. We end up generating 10,000 text-generated videos.

### 3.3 Subjective Study

To obtain the MOS of each video, we invite 27 subjects to score the perceptual quality of each video. The subjects are asked to score mainly from two aspects, in terms of text-video alignment and video fidelity. Text-video alignment refers to how the generated video content matches the text description. Video fidelity refers to degrees of distortion, saturation, motion consistency, and content rationality. The subjects use a slider ranging from 0 to 100 to give the final score of each video. After having the raw MOS of each subject, we conduct normalization to avoid inter-subject scoring differences as Z-score MOS (MOSz). That is:

$$MOSz_i = \frac{1}{N}\sum_{j=1}^{N} Res\left(\frac{r_{ij} - \mu_j}{\sigma_j}\right), \qquad (1)$$

where $i$ and $j$ refer to the index of videos and subjects. $r$ is the raw score, and $\mu_j = \frac{1}{M}\sum_{i=1}^{M} r_{ij}$, $\sigma_j = \sqrt{\frac{1}{M-1}\sum_{i=1}^{M}(r_{ij} - \mu_j)^2}$. $M$ is the number of videos scored by each subject. $N$ is the number of scores on one video. $Res(\cdot)$ is the rescaling function, converting the

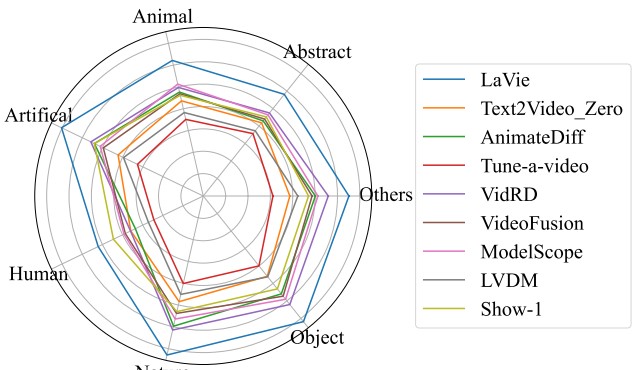

Figure 4: Comparison of models performance on different prompt types.

distribution of Z-scores into a mean of 50 and a standard deviation of 16.6. Fig. 3b shows the distributions of raw MOS and Z-score MOS.

### 3.4 Dataset Analysis

We also conduct comprehensive experiments and analysis on T2QA-DB. We first investigate each model's performance in T2QA-DB. The visualization of the Z-score MOS distributions of 10 models is shown in Fig. 3c. LaVie has the highest average MOS of 66.9, while two Tune-a-Video models have the lowest of 39.1 and 39.9. The reason for the poor performance of Tune-a-Video is mainly the low inter-frame consistency, as shown in Fig. 7c.

Based on the prompt contents, we classify the collected prompts into 6 categories, including nature, human, artificial, animal, object, and abstract. The ones that cannot be categorized into the 6 classes are labeled as "others". After classification, we have 327 prompts for nature, 119 for human, 196 for artificial, 121 for animal, 66 for object, 135 for abstract, and 36 for others. Subsequently, we compare the models' performance over different prompt types. As shown in Fig. 4, LaVie outperforms the other models on all types of prompts. Tune-a-video has the worst performance, which is consistent with the analysis in Fig. 3c. LVDM has the second-worst performance

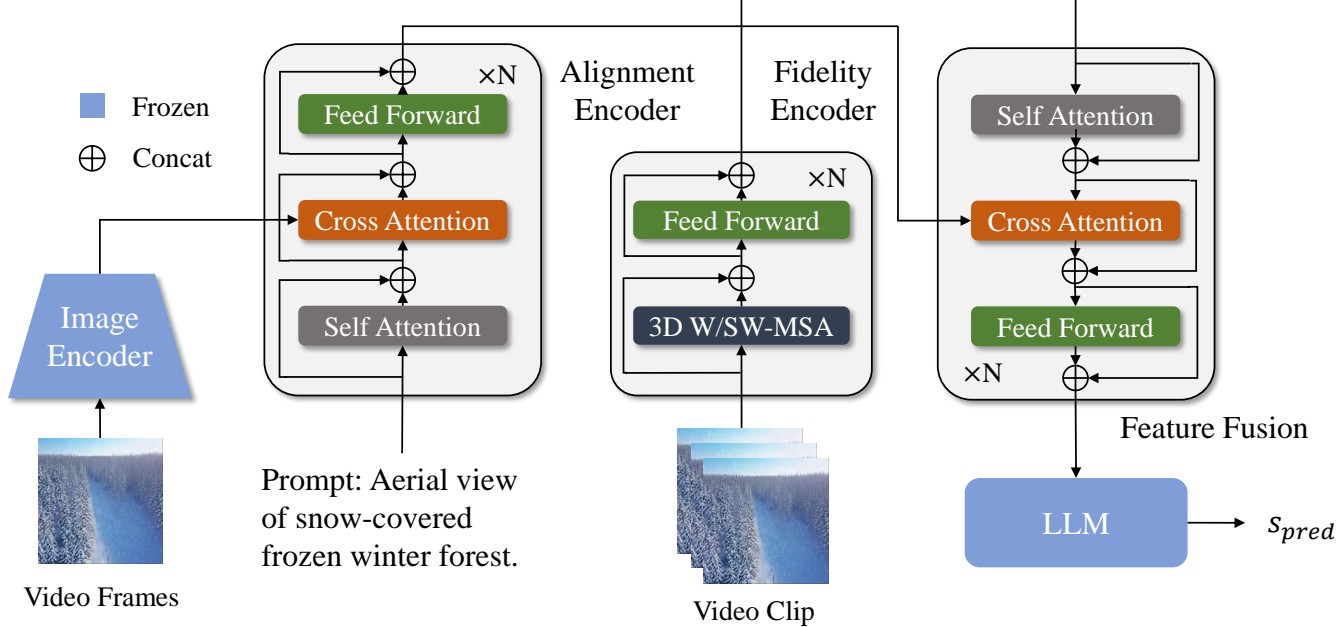

**Figure 5: Overview framework of T2VQA. Features from text-video alignment and video fidelity perspectives are extracted. After a cross-attention based fusion, an LLM is utilized for regression.**

except in the "others" type. The performance of the rest models has negligible differences. Fig. 4 also shows prompts classified as human have the worst performance in all models. The reason could be that human faces and actions require more sophisticated modeling compared with other categories. Some models show a preference for object and nature types of prompts to a small extent. The models do not show obvious preferences for other types of prompts.

## 4 SUBJECTIVE-ALIGNED TEXT-TO-VIDEO METRIC

Based on T2VQA-DB, we propose a novel model that leverages transformer-based architecture for Text-to-video Quality Assessment (T2VQA). The model dissolves the task into two perspectives, in terms of text-video alignment and video fidelity. After feature extraction and feature fusion, an LLM is used for quality regression. Fig. 5 shows the overview of the architecture of T2VQA. We will introduce the detailed design of T2VQA below.

### 4.1 Text-video Alignment Encoder

Text-video alignment refers to the conformity between the video content and the text description. CLIP [32] and BLIP [27] are models that have a strong ability for zero-shot text-image matching. [46] first proposes CLIPSim, which uses CLIP to calculate the similarities between text and each frame of the video and then take the average value. However, former works [5, 16, 28, 29] and results in Tab. 2 show that simply using CLIP or BLIP has a low correlation with the authentic subjective scores. Following works of MLLMs such as BLIP-2 [26], InstructBLIP [6], and mPLUG-Owl2 [59], we use the pre-trained BLIP image encoder as the video frame encoder. We freeze the weights of the image encoder that encodes each frame

separately. We use the BLIP text encoder as the alignment encoder. The alignment encoder takes the encoded text and video frame as inputs. They interact through the cross-attention module and the encoder eventually outputs a feature representing the text-frame similarity. We concatenate features from each text-frame pair. Given a set of $N$ video frames $\{v_i \in \mathbb{R}^{3 \times H \times W}\}_{i=1}^{N}$ and the text prompt $t$, we have:

$$f_b = cat(\{\text{BLIP}(t, v_i)\}_{i=1}^{N}). \tag{2}$$

### 4.2 Video Fidelity Encoder

Video fidelity refers to the perception of distortion from spatial and temporal domains. In the spatial domain, common distortion types include blurriness, noises, low/high contrast, etc. Temporal distortions include jitter, stall, motion blur, etc. In this perspective, the task can be seen as a common VQA task. Swin-T [30] has been proven for its excellent ability in various VQA tasks [47, 50]. By using 3D-shifted window-based multi-head self-attention (SW-MSA), Swin-T has a strong ability to analyze videos from spatial and temporal domains. Therefore, we utilize Swin-T as the backbone of the fidelity encoder to extract features that represent video fidelity. Given a video clip $v \in \mathbb{R}^{3 \times N \times H \times W}$, we have:

$$f_s = \text{SWIN}(v). \tag{3}$$

### 4.3 Feature Fusion

Inspired by BLIP-2 [26] and InstructBLIP [6], after having the features from perspectives of text-video alignment and video fidelity, we design a transformer-based fusion module to fuse those two

**Table 2: Performance of the SOTA models and T2VQA on T2VQA-DB. The best-performing model is highlighted in each column. [Bold: the best].**

| Type | Models | T2VQA-DB Validation | | | | Sora Testing | | | |
|---|---|---|---|---|---|---|---|---|---|
| | | SROCC ↑ | PLCC ↑ | KRCC ↑ | RMSE ↓ | SROCC ↑ | PLCC ↑ | KRCC ↑ | RMSE ↓ |
| zero-shot | CLIPSim [32] | 0.1047 | 0.1277 | 0.0702 | 21.683 | 0.2116 | 0.1538 | 0.1406 | 18.316 |
| | BLIP [27] | 0.1659 | 0.1860 | 0.1112 | 18.373 | 0.2126 | 0.1038 | 0.1515 | 18.850 |
| | ImageReward [58] | 0.1875 | 0.2121 | 0.1266 | 18.243 | 0.0992 | 0.0415 | 0.0748 | 19.494 |
| | ViCLIP [45] | 0.1162 | 0.1449 | 0.0781 | 21.655 | 0.2567 | 0.1844 | 0.1734 | 17.982 |
| | UMTScore [29] | 0.0676 | 0.0721 | 0.0453 | 22.559 | 0.2594 | 0.0840 | 0.1680 | 19.057 |
| finetuned | SimpleVQA [39] | 0.6275 | 0.6338 | 0.4466 | 11.163 | 0.0340 | 0.2344 | 0.0237 | 16.687 |
| | BVQA [23] | 0.7390 | 0.7486 | 0.5487 | 15.645 | 0.4235 | 0.2489 | 0.2635 | 17.164 |
| | FAST-VQA [47] | 0.7173 | 0.7295 | 0.5303 | 10.595 | 0.4301 | 0.2369 | 0.2939 | 17.426 |
| | DOVER [50] | 0.7609 | 0.7693 | 0.5704 | 9.8072 | 0.4421 | 0.2689 | 0.2757 | 17.182 |
| **Ours** | **T2VQA** | **0.7965** | **0.8066** | **0.6058** | **9.0221** | **0.6485** | **0.3124** | **0.4874** | **16.511** |

features. The module includes $N$ blocks with self-attention, cross-attention, and feed-forward layers in each block. The fidelity feature $f_s$ first goes through self-attention layers, and then it interacts with the alignment feature $f_b$ in cross-attention layers (and every other transformer block). The fusion module helps the model to unify the features from two perspectives to have a more comprehensive understanding of the video characteristics. We initialize the fusion module using $BERT_{base}$ [19].

### 4.4 Quality Regression

LLMs have been proven to be competitive on IQA/VQA tasks [52–54, 64]. Inspired by them, we also utilize an LLM as the quality regression module in T2VQA. We first design a text instruction prompt, *i.e.*, "*Please rate the quality of this video.*", to guide the LLM. The encoded instruction and the fused feature are concatenated as the input of the LLM. Following [53, 61], we supervise the LLM to output one among five ITU-standard [17] levels (*bad, poor, fair, good,* and *excellent*) to represent the quality of the videos, denoted as <level>. We assign them weights of $1-5$ in order. Since the logit at <level> in LLM is the probability distribution of all tokens, it can be used to represent how the LLM predicts the quality of the video. Therefore, a softmax for each token is calculated and multiplied by its weight. We have the final predicted score as:

$$s_{pred} = \sum_{i=1}^{5} i \times softmax(\lambda_i) = \sum_{i=1}^{5} i \times \frac{e^{\lambda_i}}{\sum_{j=1}^{5} e^{\lambda_j}}, \quad (4)$$

where $\lambda_i$ is the probability distribution of the i-th <level> token.

## 5 EXPERIMENTS AND RESULTS

### 5.1 Implement Details

*5.1.1 Train-test splitting.* When training and testing on T2VQA-DB, we follow the common practice of dataset splitting by leaving out 80% for training, and 20% for testing. To eliminate the bias in one single split, we randomly split the dataset 10 times, and use the average results for performance comparison.

*5.1.2 Training Settings.* We utilize the large model of BLIP for the extraction of text-video alignment feature. We initialize the fidelity encoder using Swin-T pre-trained on Kinetics-400 [18] dataset. For the LLM, we use the 7B model of Vicuna v1.5 [65], which is fine-tuned from Llama 2 [41]. It is worth noting that both BLIP image encoder and LLM are frozen during training.

During training and testing, we first uniformly sample 8 frames out of an input video, and then we resize them to $224 \times 224$. We use Adam optimizer initialized by a learning rate of $1e-5$. The learning rate decays under a cosine scheduler from 1 to 0. We train T2VQA for 30 epochs under a batch size of 4 on a server with one NVIDIA GeForce RTX 4090.

*5.1.3 Loss Function.* Following [47], we use differentiable Pearson Linear Correlation Coefficient (PLCC) and rank loss as loss functions. PLCC is a common criterion used for evaluating the correlation between sequences, while rank loss is introduced to help the model distinguish the relative quality of videos better. The final loss function is defined as:

$$L = L_{plcc} + \lambda \cdot L_{rank}, \quad (5)$$

where $\lambda$ is a hyper-parameter for balancing, and is set to 0.3 during training.

*5.1.4 Evaluation Metrics.* Along with PLCC, we include Spearman's Rank-Order Correlation Coefficient (SROCC), Kendall's Rank-order Correlation Coefficient (KRCC), and Root Mean Square Error (RMSE) as performance criteria. SROCC, PLCC, and KRCC indicate the prediction monotonicity, while RMSE measures the prediction accuracy. Better models should have larger SROCC, KRCC, and PLCC scores, but conversely for RMSE. Before calculating PLCC, we follow the same procedure in [10] to map the objective score to the subject score using a four-parameter logistic function.

### 5.2 Performance Comparison

*5.2.1 Reference Algorithms.* We use CLIPSim [46], BLIP [27], ImageReward [58], ViCLIP [45], UMTScore [29], SimpleVQA [39],

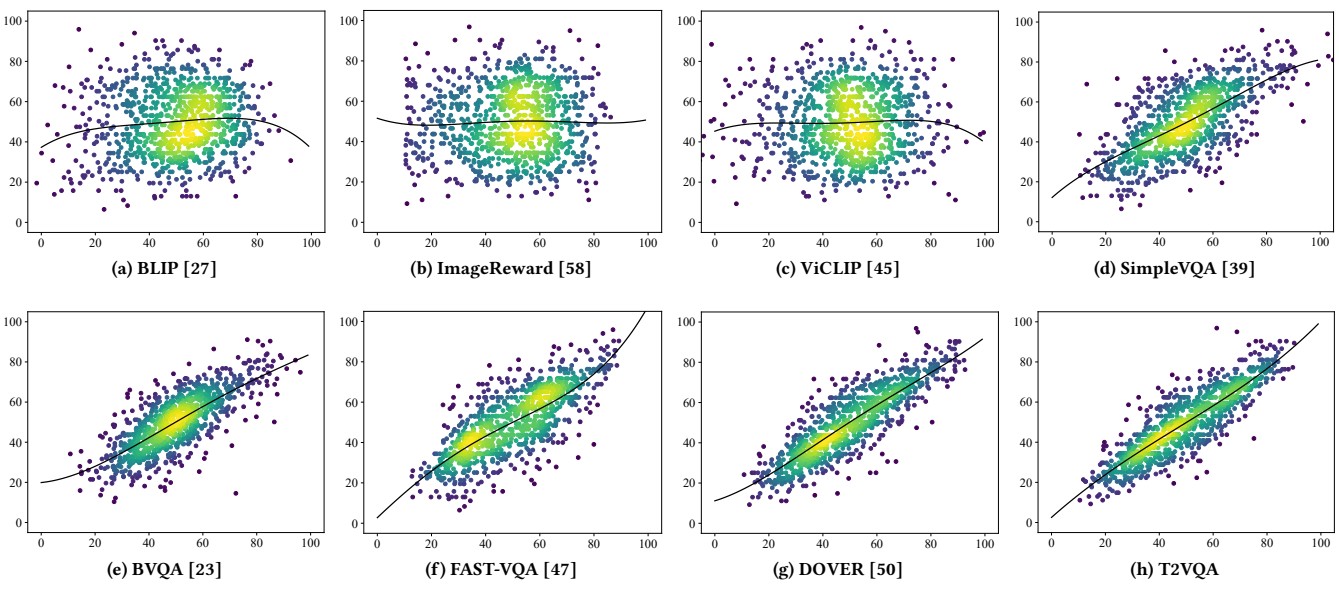

**Figure 6: Scatter plots of the predicted scores vs. MOSs. The curves are obtained by a four-order polynomial nonlinear fitting. The brightness of scatter points from dark to bright means density from low to high.**

BVQA [23], FAST-VQA [47], and DOVER [50] as the reference algorithms. DOVER is considered the SOTA VQA method to date. CLIPSim averages the similarity values between the text and each video frame. We adopt the same operation on BLIP and ImageReward to tune IQA metrics into VQA metrics. ViCLIP and UMTScore are metrics designed for measuring text-video alignment. SimpleVQA, BVQA, FAST-VQA, and DOVER are models designed for general VQA tasks. We use the pre-trained weights of CLIPSim, BLIP, and ImageReward for zero-shot testing, and we finetune SimpleVQA, BVQA, FAST-VQA, and DOVER on T2VQA-DB. All results are averaged after ten-fold splitting.

*5.2.2 Results Analysis.* Tab. 2 lists the performance comparison between T2VQA and other SOTA models. Results show that T2VQA performs best in SROCC, surpassing the second by 4.68% in SROCC and 4.85% in PLCC. The zero-shot models all have relatively low scores. They either only consider the text-video alignment or don't analyze the temporal domain information within video frames. The VQA models have higher scores, indicating that video fidelity heavily affects the assessment of text-generated video quality. However, a single perspective from video fidelity cannot address the problem properly, as there are circumstances where a high-fidelity video is generated but does not match the prompt.

Fig. 6 shows scatter plots between the predicted scores and the Z-Score MOS of 8 models. We randomly sample 1,000 videos from T2VQA-DB for testing. A better model should have a fitted curve close to the diagonal and have less dispersed scatter points. As shown in Fig. 6, T2VQA also outperforms the others.

*5.2.3 Cross-dataset Validation on Sora.* Sora [3] has been considered the SOTA T2V generation model since its release. Since the datasets listed in Tab. 1 haven't released their subjective scores, we collect the videos generated by Sora To validate the generalization

of T2VQA and other models. We collect in total of 48 videos from the official website of Sora. We invite 20 annotators to score the quality of each video. We have all the reference models tested on the Sora videos. T2VQA and other reference VQA models are trained on T2VQA-DB and tested on Sora videos. We report the SROCC, PLCC, KRCC, and RMSE between the model predictions and the ground truth MOSs. The results are listed in Tab. 2.

Experimental results show that T2VQA has the best ability of generalization among all models. Noticed that there is a performance drop between the validation on T2VQA-DB and the testing on Sora. That's because the Sora videos have the attributes of high resolution, high frame rate, and long length, which the other current T2V models are not able to generate. We will include Sora generated videos in T2VQA-DB in future work.

*5.2.4 Qualitative Analysis.* We also conduct a qualitative analysis on three examples with good, fair, and poor quality. We use SimpleVQA [39], BVQA [23], FAST-VQA [47], DOVER [50], and T2VQA to predict their quality. Fig. 7 presents the prompts, video frames, and model predictions on the three examples, and Tab. 3 lists the models' predictions and MOSs. Results show that T2VQA has more subjective-aligned predictions. Fig. 7a shows an example with a relatively high score. The scene in the video matches the description in the prompt well, and the video frames are clear and consistent. Fig. 7b is a medium-level example. Though the video matches the prompt basically, the video frames suffer from blurriness, which is reflected in the MOS and T2VQA's prediction. Fig. 7c shows the worst case among the three examples. The video fails to accurately present the description in the prompt. Besides, it loses consistency between video frames. Although it has a high definition in each frame separately, it still has low scores in both MOS and T2VQA's prediction.

**Table 3: T2VQA and other models predictions on 3 example videos. Colored numbers represent the distance between predictions and the ground truths. Red: the best.**

| Model | Prompt 1 | Prompt 2 | Prompt 3 |
|-------|----------|----------|----------|
| SimpleVQA [39] | 72.89$_{-5.33}$ | 61.59$_{+3.88}$ | 49.95$_{+17.39}$ |
| BVQA [23] | 73.12$_{-5.1}$ | 60.37$_{+2.66}$ | 45.68$_{+13.12}$ |
| FAST-VQA [47] | 85.28$_{+7.06}$ | 44.19$_{-13.52}$ | 38.91$_{+6.35}$ |
| DOVER [50] | 89.11$_{+10.89}$ | 51.93$_{-5.78}$ | 36.66$_{+4.1}$ |
| **T2VQA(Ours)** | 81.61$_{+3.39}$ | 57.11$_{-0.6}$ | 29.07$_{-3.49}$ |
| MOS (gt) | 78.22 | 57.71 | 32.56 |

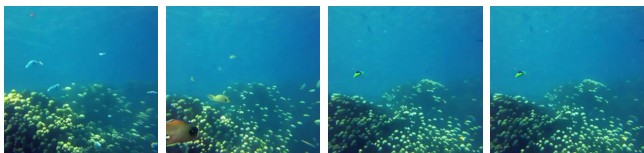

**(a) Prompt: Castle ruins on the hill in the middle of a beautiful landscape.**

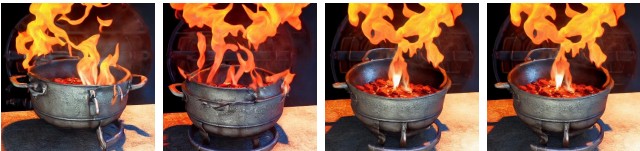

**(b) Prompt: Underwater world with different fishes, corals, and stones.**

**(c) Prompt: In a hot cast-iron cauldron, the cook pours oil to fry the meat (liver).**

**Figure 7: Three video examples with good, fair, and poor quality.**

## 5.3 Ablation Studies

To validate the effectiveness of each module in T2VQA, we conduct thorough ablation studies, including the alignment and fidelity encoder, the fusion module, and the regression module. Experimental results are listed in Tab. 4. All results are averaged after 10-fold splitting.

*5.3.1 Alignment Encoder.* T2VQA utilizes the BLIP image encoder and text encoder as the alignment encoder. CLIP is another widely used text-image encoder. It has a similar structure and ability to measure text-video alignment as BLIP. We replace the BLIP image and text encoder in T2VQA with CLIP to determine which one has the better performance.

Experimental results show that using CLIP as the alignment encoder suffers from severe performance degradation. The reason could be that the image and the text are encoded separately in CLIP.

**Table 4: Performance comparison of ablation studies. [Keys: Bold: the best].**

| Models | Validation | | | |
|--------|------------|------|------|------|
| | SROCC ↑ | PLCC ↑ | KRCC ↑ | RMSE ↓ |
| T2VQA-CLIP | 0.7296 | 0.7347 | 0.5385 | 10.5141 |
| T2VQA-resnet | 0.7610 | 0.7730 | 0.5715 | 9.8152 |
| T2VQA-cat | 0.7734 | 0.7854 | 0.5839 | 9.4034 |
| T2VQA-linear | 0.7808 | 0.7919 | 0.5891 | 9.3011 |
| T2VQA-nonlinear | 0.7850 | 0.7983 | 0.5954 | 9.1755 |
| **T2VQA(Ours)** | **0.7965** | **0.8066** | **0.6058** | **9.0221** |

While in BLIP, the encoded image and the text features interact in the cross-attention module.

*5.3.2 Fidelity Encoder.* In T2VQA, we use the Swin-T as the backbone of the fidelity encoder. To investigate the effectiveness of the transformer-based architecture, we conduct the control experiment by using the convolution-based 3D ResNet [13] as the fidelity encoder. The results show that Swin-T has a superior performance to ResNet, validating the effectiveness of the transformer-based architecture.

*5.3.3 Fusion and Regression Modules.* Besides, we compare our cross-attention fusion strategy with the simple concatenation fusion. The latter is the simplest yet most commonly used fusion strategy. In T2VQA, we take advantage of the strong ability of an LLM for quality regression. We test the commonly used linear regression and non-linear regression to compare with the LLM regression in T2VQA. For linear regression, we use two full-connected layers with 128 neurons in the first layer and 1 neuron in the second. For non-linear regression, we use two 1D convolution blocks with kernel size set to 1. We also set the channel number to 128 in the first block and 1 in the second.

Results in Tab. 4 show that T2VQA achieves the best performance in all evaluating metrics, indicating its effectiveness. The models using concatenation, linear regression, and non-linear regression have similar performance, yet they are all inferior to T2VQA, indicating that the cross-attention fusion and LLM have achieved non-negligible improvement to the model.

## 6 CONCLUSION

In conclusion, in this paper, we are dedicated to giving a subjective-aligned prediction of the quality of a text-generated video. For that purpose, we establish a T2V dataset with the largest scale, named T2VQA-DB. The dataset includes 10,000 videos generated by 9 advanced T2V models. We also conduct a subjective study to obtain the MOSs on the overall video quality. Based on T2VQA-DB, we propose a novel transformer-based model for text-to-video quality assessment, named T2VQA. The model extracts features of a video from text-video alignment and video fidelity perspectives respectively. After fusing the features, an LLM is utilized to regress the final prediction. The experimental results indicate that T2VQA is effective in evaluating the quality of text-generated videos.

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
