# OpenReview forum: "Subjective-Aligned Dataset and Metric for Text-to-Video Quality Assessment"
_acmmm.org/ACMMM/2024/Conference — MM2024 Oral_

### Official Review · Reviewer_Qfse · 2024-05-07

**Rating:** 6
**Confidence:** 4

**Summary:**

The paper presents a dataset for text-to-video generation and a model to estimate the quality of such ai generated videos.
For me, it is somehow unclear whether the dataset or the models are published as open source/data.
This would be a really good addition to the field.
Overall the paper is good and aims at a highly interesting topic.

Note: the dataset has been used already in a competition https://paperswithcode.com/paper/ntire-2024-quality-assessment-of-ai-generated. Furthermore the paper has been uploaded already to arxiv https://arxiv.org/abs/2403.11956 (with clear names of the authors)

# detailed comments
## abstract

* will the videos and or model being released? if so, then please make this clear in the abstract

## introduction

* fig.1 is never referenced in the text?
* "Fig. 2 shows video examples from T2VQA-DB", explain what is shown in a) and what in b) in the text, as addition, also please name the used 9 T2V generators in the text (and not only in the Fig), why have they being selected?
* Moreover, Fig2, b) seems to show "the videos", so these videos are short term 4 frame videos?, thus ratheer GIFs style videos? which durations have the generated videos? which resolution?
* so 27 subjects rated 10.000 videos? how long did this took?

## related work

* 2.1: for the datasets it would be also interesting to know which resolution, framerate, and duration the videos have, especially considering "quality".
* furthermore, maybe also linking to "traditional video quality" datasets, such as KoNViD-1k (https://database.mmsp-kn.de/konvid-1k-database.html )

## subjective aligned text to video dataset
* the intro text can be compressed, this text was already more or less used in the abstract and in the intro
* 3.1. the selection of the prompts seems to be purely data driven, however was there any "inspection" of the resulting text prompts? why were no "genres" used for the filtering
* 3.2. how are the videos stored? in a compressed format or in a lossless format? (would be important for a quality analysis)

* 3.3. "Z-scores into a mean of 50 and a standard deviation
of 16.6" where are theses numbers comming from?
    * as I get it from the equaiton, not every subject rated all videos? how many rated?
    * how long was the test?
    * which screen? was it a test lab environment?
    * (check some ITU Recommendations for subjective testing, e.g. ITU P.910)
* so the data will not be shared?, so far I did not got any comment about sharing in this text

## subjective aligned t2v metric
* eq. 2, what is `cat` doing? (add a note in the text, `cat` can have different meanings)
* eq.4 I'm not sure if it is needed to add the full form of the softmax part in the right side of the equation

## experiments and results
* "DOVER is considered the SOTA VQA method to date.", Or Q-Align? what about this, it was mentioned in the SoA but not used in the eval? it is also based on LLM, thus maybe more comparable to the proposed model?
* 5.2.2. "SROCC, surpassing the second by 4.68% in SROCC
and 4.85% in PLCC." is this for the proposed dataset or the Sora dataset? (right side of the table), please clarify
* "We randomly sample 1,000 videos from
T2VQA-DB for testing", but you did a 80-20 split for train validation, thus 10k videos -- 20% for validation, thus, 2k remaining, thus from this you sampled 1000 or how?, is there any overlapp of train and validation here?

* 5.2.3. What is Sora, it comes now, with no introduction
* "T2VQA and other reference VQA models are trained on T2VQA-DB", so you re-trained all other models using the proposed dataset? so far, I just got that your model was specifically trained for the dataset.

**Strengths:**

A large-scale database and objective models for text-to-video generated content are required for research because most of the models are either not capable of handling AI-generated content, or the available datasets are small and do not cover a wide range of generators.

**Limitations:**

Detailed comments are listed in the "Summary", in some places a clearer reasoning why specific choices (e.g. selection of generators, selection of text prompts) have been made would be good to have.
Furthermore, details to the subjective studies (which screen, was it in a controlled lab environment, is it following any kind of ITU Rec. and so on) would be good to have.
Especially the rating of 10k videos from 27 subjects is confusing, and it is not clear for me in the text if a subject rated all videos or only a part, was there any payment? how long was such a test?

**Suitability:**

3

---

### Official Review · Reviewer_WEY2 · 2024-05-09

**Rating:** 4
**Confidence:** 4

**Summary:**

This paper proposes to evaluate the quality of text-generated videos quantitatively. To this end, the authors establish T2VQA-DB, a large-scale text-generated video dataset. Based on T2VQA-DB, they also propose T2VQA, a novel transformer-based model for T2V quality assessment. Experimental results show that T2VQA outperforms existing T2V metrics and SOTA video quality assessment models.

**Strengths:**

1.	The language used in the paper has good clarity and is easy to follow, and the ideas are presented well through illustrations and experimental results.
2.	The paper builds the largest-scale T2V dataset with mean opinion scores (10,000 videos, 9 different T2V models). It will facilitate the development of more T2V metrics and T2V generation models.
3.	The analysis of the dataset is comprehensive. It demonstrates the current real application scenarios and performance status of T2V models and analyses cases where T2V models have bad performance, facilitating future T2V generation.
4.	The proposed T2VQA model creatively extracts features from text-video alignment and video fidelity perspectives. It fills the gap in overall text-generated video quality assessment and outperforms other SOTA methods， having a certain guidance role to future T2V generation research.

**Limitations:**

1.	Why not use the existing T2V datasets, and what is the necessity and significance for the authors to propose new datasets?
2.	There is a lack of explanation of the main technical details of Figure 5, and it is necessary to highlight the novelty of the framework proposed by the authors and its unique design for text-to-video quality assessment.
3.	The model’s generalization ability remains to be seen due to the lack of cross-dataset validation.
4.	Videos with high resolution and frame rates should be included in the dataset, making the dataset more representative.

**Suitability:**

3

---

### Official Review · Reviewer_Q5fu · 2024-05-25

**Rating:** 5
**Confidence:** 3

**Summary:**

The paper proposes a large-scale Text-to-Video Quality Assessment DataBase (T2VQA-DB) and a novel transformer-based model named T2VQA for evaluating the quality of text-generated videos. The main points and contributions are:

1:Largest T2V Dataset: The paper introduces T2VQA-DB, the largest Text-to-Video quality assessment dataset to date, containing 10,000 videos generated by 9 different T2V models.

2:Subjective Study: A subjective study is conducted with 27 subjects to obtain Mean Opinion Scores (MOS) for each video in T2VQA-DB.

3:Novel Model: A novel transformer-based model, T2VQA, is proposed for text-to-video quality assessment. The model extracts features from text-video alignment and video fidelity perspectives and leverages a large language model for quality prediction.

4:Effective Performance: Experimental results show that T2VQA outperforms existing T2V generation metrics and state-of-the-art video quality assessment models, indicating its effectiveness in evaluating text-generated videos.

5:Practical Application Prospects: The proposed T2VQA model can benefit in measuring the performance of T2V generation algorithms, giving it practical application prospects.

**Strengths:**

(1) The T2VQA-DB dataset established by the paper is of unprecedented scale, which will significantly contribute to the development of more comprehensive and accurate metrics for text-generated videos.

(2) The proposed T2VQA model takes into account both text-video alignment and video fidelity perspectives, which is a significant improvement over existing methods that only focus on one aspect.

(3) The experimental results demonstrate the effectiveness of the proposed model, which achieves state-of-the-art performance on the newly established T2VQA-DB dataset as well as another dataset.

(4) The paper provides thorough ablation studies to validate the contribution of each component in the proposed model.

**Limitations:**

(1) The implementation details of the proposed model, such as the specific hyperparameters and training procedures, are not fully disclosed, which may hinder the reproducibility of the results.

(2) The evaluation of the model is limited to two datasets. More extensive evaluations on diverse datasets would strengthen the generalizability of the results.

(3) The paper lacks a discussion of potential limitations and future directions of the work, which could provide valuable insights for readers.

**Suitability:**

2

---

### Meta-Review · Area_Chair_zBon · 2024-07-04

**Recommendation:** Accept (Oral)
**Confidence:** 4

**Metareview:**

A text-to-video quality assessment database and a transformer-based approach to evaluate the quality of text-generated videos is presented. The reviewers largely agree that the proposed method is interesting and novel, and the paper is unique in several aspects: integration of text-video alignment and video fidelity perspectives, the performance of the proposed approach, and the large size of the dataset. Furthermore, the authors reacted to mentioned limitations by the reviewers and outlined appropriate changes to be made in the final version of the paper (which need to be checked by the program chairs before publication).